# Studying the Roles of the Renin–Angiotensin System in Accelerating the Disease of High-Fat-Diet-Induced Diabetic Nephropathy in a db/db and ACE2 Double-Gene-Knockout Mouse Model

**DOI:** 10.3390/ijms25010329

**Published:** 2023-12-26

**Authors:** Cheng-Yi Chen, Meng-Wei Lin, Xing-Yang Xie, Cheng-Han Lin, Chung-Wei Yang, Pei-Ching Wu, Dung-Huan Liu, Chih-Jen Wu, Chih-Sheng Lin

**Affiliations:** 1Division of Nephrology, Department of Internal Medicine, Mackay Memorial Hospital, Hsinchu 300, Taiwan; ricechen@hotmail.com; 2MacKay Junior College of Medicine, Nursing and Management, Taipei 112, Taiwan; 3Department of Biological Science and Technology, National Yang Ming Chiao Tung University, Hsinchu 300, Taiwan; lmw1018@nycu.edu.tw (M.-W.L.); josedulcimer@gmail.com (X.-Y.X.); a0975273923@gmail.com (C.-H.L.); 4Division of Nephrology, Department of Internal Medicine, National Taiwan University Hospital Hsinchu Branch, Hsinchu 300, Taiwan; yang.chung.wei@gmail.com; 5Doctoral Degree Program of Biomedical Science and Engineering, College of Biological Science and Technology, National Yang Ming Chiao Tung University, Hsinchu 300, Taiwan; d36619@mail.cmuh.org.tw (P.-C.W.); d35345@mail.cmuh.org.tw (D.-H.L.); 6Department of Chinese Medicine, China Medical University Hospital, Taichung 404, Taiwan; 7Department of Physical Medicine and Rehabilitation, China Medical University Hospital, Taichung 404, Taiwan; 8Division of Nephrology, Department of Internal Medicine, Mackay Memorial Hospital, Taipei 100, Taiwan; 9Division of Medicine, College of Medicine, Taipei Medical University, Taipei 100, Taiwan; 10Center for Intelligent Drug Systems and Smart Bio-devices (IDS2B), National Yang Ming Chiao Tung University, Hsinchu 300, Taiwan

**Keywords:** diabetic nephropathy, renin angiotensin system, high-fat-diet, angiotensin converting enzyme II (ACE2), chymase

## Abstract

Diabetic nephropathy (DN) is a crucial metabolic health problem. The renin–angiotensin system (RAS) is well known to play an important role in DN. Abnormal RAS activity can cause the over-accumulation of angiotensin II (Ang II). Angiotensin-converting enzyme inhibitor (ACEI) administration has been proposed as a therapy, but previous studies have also indicated that chymase, the enzyme that hydrolyzes angiotensin I to Ang II in an ACE-independent pathway, may play an important role in the progression of DN. Therefore, this study established a model of severe DN progression in a db/db and ACE2 KO mouse model (db and ACE2 double-gene-knockout mice) to explore the roles of RAS factors in DNA and changes in their activity after short-term (only 4 weeks) feeding of a high-fat diet (HFD) to 8-week-old mice. The results indicate that FD-fed db/db and ACE2 KO mice fed an HFD represent a good model for investigating the role of RAS in DN. An HFD promotes the activation of MAPK, including p-JNK and p-p38, as well as the RAS signaling pathway, leading to renal damage in mice. Blocking Ang II/AT1R could alleviate the progression of DN after administration of ACEI or chymase inhibitor (CI). Both ACE and chymase are highly involved in Ang II generation in HFD-induced DN; therefore, ACEI and CI are potential treatments for DN.

## 1. Introduction

Diabetes mellitus (DM), especially type 2 diabetes mellitus (T2DM), is a major public-health problem worldwide [1]. Fueled by the global rise in the prevalence of obesity and unhealthy lifestyles, T2DM has become one of the fastest-growing diseases [2]. T2DM is related to a high prevalence of diabetic nephropathy (DN): 20–40% of DM patients with microalbuminuria progress to manifest nephropathy [3,4]. In T2DM research, mice with genetic defects in the leptin receptor (db/db) have been widely used as models [5]. Because these animal models enable the development of innovative experimental strategies that are not suitable for clinical studies, they are useful for elucidating the pathogenesis of human disorders, and mice are particularly useful. Due to their differential susceptibility to DM and renal diseases, mice have become one of the main experimental models of choice for preclinical studies of human disease in many fields, including DM and DN research [6,7]. The requisite and most important step in using mice as a DN model is to generate insulin resistance and/or chronic hyperglycemia [8]. It is known that feeding a high-fat diet (HFD) can induce obesity, hyperlipidemia, insulin resistance, hyperglycemia, chronic inflammation and high oxidative stress in mice [9]. Thus, it is possible to model more serious DN that is similar to advanced human disease, including progressive elevations in albuminuria and serum creatinine and pathological changes. The model involves both genetic defects and feeding an HFD [10].

The renin–angiotensin system (RAS) is a classic hormonal system that is important in homeostasis in the circulation and tissues, especially in the cardiovascular and renal systems. Angiotensin-converting enzyme (ACE) is the primary enzyme that converts angiotensin I (Ang I) to angiotensin II (Ang II) in the RAS [11]. Unbalanced RAS activity and an abnormally activated ACE/Ang II axis are the major effectors that may contribute to the onset and progression of renal damage (Figure 1) [12,13]. Excessive local concentrations of Ang II may also contribute directly, accelerating renal damage by promoting cell growth, inflammation, and fibrosis [14,15]. Clinical therapy for renal diseases usually uses angiotensin-converting enzyme inhibitor (ACEI) and Ang II receptor blockers (ARBs) to decrease ACE/Ang II activation and ameliorate disease development. However, treatment results vary across people and disease states. Even with the combinational use of ACEI and ARBs, the effectiveness of this treatment is still subject to debate [16,17].

The need for new therapeutic targets is now fueled by the failures of traditional RAS blockers, such as direct renin inhibitor, aliskiren, and chymase inhibitor [18,19]. Chymase is a serine protease secreted by granules of mast cells, which produce a large array of inflammatory mediators [20]. A well-known function of chymase is converting Ang I to Ang II in an ACE-independent pathway. The catalytic activity of chymase in the conversion of Ang I to Ang II is about 20-fold higher than that of ACE [21]. Due to the higher catalytic activity of Ang II compared to ACE, various studies have been conducted on the chymase-dependent formation of Ang II [22,23]. Multiple reports have shown that human mast-cell chymase has potential as a new drug target [23,24]. Multiple lines of evidence have suggested alternative pathways to ACE conversion for the generation of Ang II in the heart and kidneys. There has been growing interest in the role of chymase in various renal pathophysiologic states. Increased chymase expression has also been observed in humans with DN [25,26].

Additionally, angiotensin-converting enzyme II (ACE2), which was first cloned as a homolog of human ACE by two independent research groups in 2000 [27,28], cleaves the C-terminal amino acid of Ang II to generate the peptide angiotensin 1–7 (Ang 1–7). In RAS, ACE2/Ang 1–7 is thought to counteract the function of the ACE/Ang II axis and may have a function in tissue protection (Figure 1) [12,13]. ACE2 is known to be present in human kidneys, but little data is available regarding its role in renal disease [29]. In the kidneys, ACE2 is largely localized in glomerular mesangial cells and tubular epithelial cells, and the altered expression and function of ACE2 are associated with hypertensive renal disease and diabetic kidney disease in multiple models of kidney injury [30]. Much of the evidence significantly supports the importance of the ACE2-Ang 1–7/Mas axis in kidney function. Decreased glomerular ACE2 expression is found in the diabetic kidney, and this change may increase intraglomerular Ang II and contribute to aggravating kidney injury [30,31].

The pathological mechanisms of DN are intricate and complex. Currently, there is no animal model that can be used to simulate all the pathological features of human DN, especially the pathological changes characteristic of the progressive and late stages of DN. Thus, the purposes of this study are to explore DN progression in db/db and ACE2 KO mice and elucidate the role of RAS factors in DN and changes in their activity after HFD feeding. That is, a genetic deficiency in ACE2 was paired with short-term HFD feeding in db/db mice to accelerate DN progression in this study. Besides the general ACE inhibitor (captopril), a chymase inhibitor (chymostatin) was also given to the mice during HFD feeding. This study seeks to compare the effects of these two drugs, investigate the kidney injury resulting from DN via biochemical analysis, detect protein expression levels, and analyze tissue lesions that occur in the kidney during DN development.

**Figure 1 ijms-25-00329-f001:**
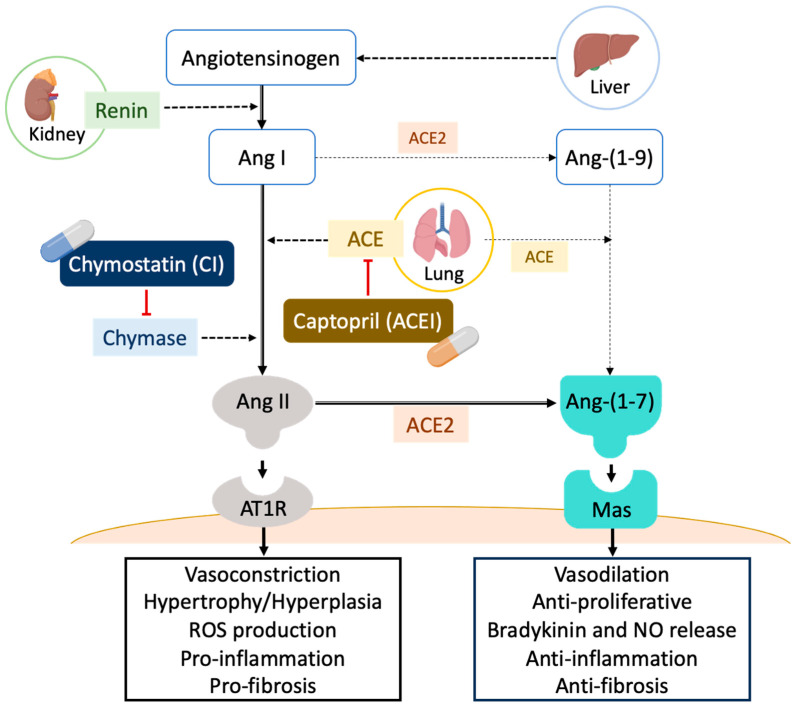
Schematic representation of RAS cascade. The renin–angiotensin system (RAS) is a major regulatory system involved in blood pressure and water balance. The classical RAS is made up of a circulating endocrine system in which renin cleaves angiotensin I (Ang I). Inactive Ang I is then hydrolyzed by angiotensin-converting enzyme (ACE) and also by chymase, producing the octapeptide angiotensin II (Ang II). Ang II then binds to Ang II Type-I receptors (AT1R), which causes vasoconstriction and inflammation. Angiotensin-converting enzyme II (ACE2) cleaves the Ang II to generate the peptide angiotensin 1–7 (Ang 1–7), which acts via the Mas receptor, i.e., as Ang 1–7/Mas, to counteract the adverse effects of Ang II/AT1R. ACE and chyamse activity can be inhibited by an ACE inhibitor (ACEI; Captopril) and a chymase inhibitor (CI; Chymostatin), respectively [13,32].

## 2. Results

### 2.1. Changes in Physical Characteristics

Thirty male db/db and ACE2 KO mice at seven weeks of age were used in this study. After one week of acclimatization, all mice were randomly divided into five groups. These groups were as follows: (1) 8 weeks old, ND; (2) 12 weeks old, ND; (3) 12 weeks old, HFD; (4) 12 weeks old, HFD + captopril (ACEI); and (5) 12 weeks old, HFD + chymostatin (chymase inhibitor; CI) (Figure 2).

The results shown in Table 1 indicate the body weight and blood glucose of the db/db and ACE2 KO mice measured in each group each week. The changes in animal body weight (g) and kidney weight/body weight (KW/BW, %) leveled out and remained steady. The measures of body weight and KW/BW in the final week in the 12-week-old ND group were 110% and 117% of their values in the first week, i.e., in the 8-week-old mice. Conversely, the mouse blood glucose and kidney weight in the 12-week-old ND group were increased significantly compared to those in the 8-week-old ND group (368 ± 20.6 vs. 256 ± 24.1 mg/dL, *p* < 0.001). In the comparisons between 12-week-old HFD and 12-week-old ND groups, the body weight, blood glucose and kidney weight were significantly greater in the 12-week-old HFD group. After four weeks of the HFD (12-week-old HFD), the body weight, blood glucose and kidney weight of mice dramatically increased to 125%, 168% and 103%, respectively, of the values in the 12-week-old ND group (Table 1). These results indicate that the HFD consistently accelerated the progression of obesity and hyperglycemia.

The administration of captopril (ACEI) did slightly change the body weight and the fasting blood glucose (*p* > 0.05). Notably, chymostatin (CI) treatment significantly stabilized the abnormal increase in fasting blood glucose from 620 ± 41.5 mg/dL to 499 ± 42.9 mg/dL (*p* < 0.001) (Table 1). Both captopril and chymostatin treatments significantly decreased the HFD-induced increase in kidney weight. The effect was more significant in the chymostatin group (i.e., 12-week-old HFD + chymostatin group). The KW/BW ratio was significantly more reduced by chymostatin treatment compared to the captopril treatment (Table 1).

### 2.2. Changes in Biochemical Characteristics of Blood and Urine

The results of the urinary biochemical assays are shown in Figure 3A–C. The blood cholesterol and triglyceride levels in the 12-week-old HFD mice were slightly elevated compared with those in the 8-week-old ND mice, but the differences did not reach statistical significance (*p* > 0.05). HFD feeding was associated with marked increases in the blood cholesterol and triglyceride levels in experimental db/db and ACE2 KO mice (*p* < 0.001 for cholesterol and *p* < 0.01 for triglyceride levels). The change in serum creatinine in the 12-week-old HFD mice was also significantly greater than that in the 8-week-old and 12-week-old ND mice (*p* < 0.001). These results suggest that an HFD for the db/db and ACE2 KO mice rapidly accelerated the progression of hyperlipidemia. Administration of captopril significantly reduced the blood triglyceride level, but not the cholesterol level, in 12-week-old HFD + ACEI mice. However, chymostatin treatment (12-week-old HFD + CI mice) significantly reduced both the cholesterol (250 ± 22 vs. 334 ± 34 mg/dL, *p* < 0.01) and triglyceride (86 ± 9 vs. 140 ± 23 mg/dL, *p* < 0.05) levels induced by HFD feeding. The reduction in serum creatinine caused by captopril and chymostatin treatments was also highly significant (*p* < 0.01). The levels of serum creatinine in the 12-week-old HFD group (0.31 mg/dL) were approximately 2.5-fold and 1.8-fold higher than in the ND-fed group at 8 and 12 weeks old, respectively. The results suggest that regulating RAS could affect lipid metabolism and rescue liver and kidney function. Decreasing the abnormal generation of Ang II could alleviate the pathogenic process of hyperlipidemia (Figure 3A–C).

Figure 3D–G shows the results of urinary biochemical assays. The trends in changes in the urinary parameter were similar to the changes seen in the physical and blood parameters. The urine creatinine levels were identical in all mouse groups, whereas the urine micro-albumin (microALB) and urine albumin–creatinine ratio (UACR) were significantly increased to 23.7 mg/dL and 143.9 mg/g, respectively, (*p* < 0.001) in the 12-week-old HFD group compared with the values in the 8-week-old and 12-week-old ND groups. As expected, the creatinine clearance rate (CCr) in the mice fed the HFD was significantly decreased to 0.06 ± 0.02 mL/min compared to 0.24–0.26 mL/min (*p* < 0.001) in the mice fed the ND. According to the results for microALB, UACR, and CCr, both captopril and chymostatin treatments significantly improved the function of kidneys damaged by the HFD in db/db and ACE2 KO mice.

### 2.3. Diabetic Nephropathy Trigged by MAPK and RAS Activation via Abnormal Production of Ang II 

Activation of the three signaling proteins of the MAPK pathway—p-ERK1/2, p-JNK and p-p38—was detected, and the results are shown in Figure 4A. The relative p-ERK1/2 expression levels were similar in all groups. The p-JNK expression levels were almost undetectable in the mice with ND supplementation but were significantly increased in the HFD-fed mice. This dramatic elevation in p-JNK expression was significantly inhibited when the HFD mice were treated with captopril or chymostatin (*p* < 0.001). Relative p-p38 expression increased with both age and HFD. Chymostatin, but not captopril, treatment significantly reduced the HFD-induced increase in p-p38 expression (*p* < 0.01) (Figure 4).

As expected, captopril treatment significantly reduced ACE expression (compared to that of untreated HFD mice) to about 2.3-fold the levels seen in the 8-week-old ND group, but chymostatin treatment increased the relative ACE expression levels by about 4-fold compared with 8-week-old ND group (Figure 5). Similar results were obtained for the relative expression levels of chymase and Ang II; chymostatin treatment significantly reduced the relative chymase expression level (compared to that of untreated HFD mice) to 1.4-fold the levels seen in the 8-week-old ND group, but captopril treatment increased the renal chymase expression level to approximately 2.6-fold greater than the levels in the 8-week-old ND group. Decreased relative Ang II expression levels were found in both the captopril (2-fold the level seen in the 8-week-old ND group) and chymostatin (1.5-fold the level seen in the 8-week-old ND group) treatment groups. Interestingly, combination treatment with captopril and chymostatin did not change ACE protein expression (3.2-fold compared with the 8-week-old ND group, as for untreated HFD mice) but did slightly decrease chymase protein expression (2-fold compared with the 8-week-old ND group, lower than the level seen in untreated HFD mice).

The data shown above demonstrate that the HFD caused the activation of RAS. Intrarenal Ang II could be generated by either ACE-dependent or chymase-dependent pathways in the db/db and ACE2 KO mice. Captopril or chymostatin dramatically ameliorated abnormal RAS activation by restoring normal Ang II levels.

### 2.4. Histopathology of the Kidney of db/db and ACE2 KO Mice

The kidney tissue sections were stained with three stains—hematoxylin and eosin (H&E), Masson’s trichrome (Masson’s) and Periodic Acid–Schiff (PAS) (Figure 6, Figure 7 and Figure 8). Figure 6A shows that db/db and ACE2 KO mice displayed glomerular hypertrophy in the HFD-fed groups. Additionally, varying degrees of change in lymphocytic infiltration were found in HFD mice (Figure 6B). In the glomerulus, 8- and 12-week-old ND groups showed similar degrees of infiltration (30%). The infiltration area was about 40% in the 12- week-old HFD group, which is markedly higher than those in all ND-fed groups (*p* < 0.001). Chymostatin, but not captopril, reversed the HFD-induced lymphocytic infiltration (12-week-old HFD + ACEI group, 34%; 12-week-old HFD + CI group, 30%) (*p* < 0.01).

Figure 7B presents fibrosis levels in the db/db and ACE2 KO mice; collagen accumulation in the glomerulus leveled out and remained steady, and the percentages of fibrosis area in the 8- and 12-week-old ND groups were 17% and 26%, respectively. In the tubules, the fibrosis levels of the 8-week-old and 12-week-old ND groups were about 10% and 16%, respectively. HFD feeding led to noticeably increased kidney damage in the glomerulus and tubules, in addition to a missing or ruptured brush border in the proximal tubule (Figure 7A; black arrow). The area of fibrosis was greater (39% in the glomerulus and 21% in the tubules) in the 12-week-old HFD group. Captopril and chymostatin significantly decreased collagen accumulation; in the glomerulus, the percentages of fibrosis area in 12-week-old HFD + captopril and 12-week-old HFD + chymostatin groups were 27% and 28%, respectively. Similar results were found for the tubules, where the percentages of fibrosis area were 15% and 21% in the 12-week-old HFD + captopril and 12-week-old HFD + chymostatin groups, respectively. However, captopril and chymostatin treatment did not resolve the missing brush border in the proximal tubule.

The accumulation levels of PAS-positive matrix in db/db and ACE2 KO mice are shown in Figure 8B. In the glomerulus, age slightly increased the percentage of mesangial matrix expansion—11% and 13% for the 8- and 12-week-old ND groups, respectively. In the tubules, the PAS-positive matrix level was insignificant, at 8% and 9% in the 8- and 12-week-old ND groups, respectively. HFD feeding massively enhanced the percentage of mesangial matrix expansion to about 21% in the glomerulus and 18% in the tubules in 12-week-old HFD group. Additionally, HFD feeding also resulted in severe tubular atrophy (Figure 8A; black arrow). Captopril and chymostatin treatment reduced tubular damage; the percentages of PAS-positive matrix area in tubules in the 12-week-old HFD + captopril and 12-week-old HFD + chymostatin groups were 9% and 7%, respectively. Furthermore, tubular atrophy was ameliorated in treated mice. However, captopril and chymostatin treatment did not ameliorate glomerular damage; the percentages of PAS-positive matrix area in tubules in the 12-week-old HFD + captopril and 12-week-old HFD + chymostatin groups were 24% and 23%, respectively.

## 3. Discussion

In this study, four continuous weeks of feeding an HFD in db/db and ACE2 KO mice clearly caused kidney damage proceeding to typical DN. This result could be seen in physical and biochemical changes, such as HW/KW, indicating renal hypertrophy; high blood glucose; and UACR and CCr, representing renal dysfunction. We have provided direct evidence that abnormal RAS activity can increase the progression of kidney disease when it is combined with short-term feeding of an HFD in a mouse model. The db/db and ACE2 KO mice presented severe DN at 12 weeks of age after 4 weeks on the HFD. The db/db mice, a mouse strain with a genetic leptin-receptor deficiency, can spontaneously exhibit features of T2DM. It has been reported that db/db mice can spontaneously develop hyperinsulinemia at 10 days, hyperleptinemia at 3 weeks [33], and hyperglycemia at 8 weeks [34]. The renal function of db/db mice starts declining at 15–18 weeks of age [35], and 18–20-week-old db/db mice show high levels of plasma creatinine [36,37]. We have provided an advanced animal model in which 12-week-old db/db and ACE2 KO mice quickly developed typical DN; this model could thus be used for therapeutic studies of DN. Notably, in some studies, HFD-fed db/db mice displayed similar or even more impaired organ functions, showing sharply increasing body weight, plasma blood glucose, creatinine, and blood lipid levels [38,39]. Our pathologic findings from the kidneys show that age and HFD caused the expansion of the pro-fibrotic area in the renal tubules and glomeruli (Figure 6, Figure 7 and Figure 8). The results of PAS staining show that HFD markedly increased mesangial expansion and even caused tubular atrophy [40]. DN is characterized by excessive extracellular matrix (ECM) accumulation, which leads to renal fibrosis. Matrix metalloproteinases (MMPs), as well as tissue inhibitors of metalloproteinases (TIMPs), are known to regulate synthesis and degradation of the ECM [41,42]. Kidney injury molecule-1 (KIM-1) is also markedly up-regulated in the post-ischemic kidney and in acute renal failure [43]. In the fibrotic pathogenesis of DN, KIM-1, MMPs and TIMPs play a crucial role. Their expression is triggered by oxidation and inflammatory stimulation, and their activities are being explored to clarify molecular mechanisms of fibrosis pathogenesis in the db/db and ACE2 KO mice.

ACE2 is predominantly expressed in the proximal tubule of kidneys, where it is co-localized with ACE at the luminal brush border. This tubular distribution is similar to the distribution of Ang 1–7, its major product [44]. ACE2 expression has also been documented in other parts of the nephron, including in glomerular podocytes and the endothelium of renal capillaries [45,46]. Although the angiotensinase activity of ACE2 in healthy kidneys is well understood, its role in the maintenance of normal physiology remains to be determined. Notably, both kidney function and renal development are normal in ACE2 knockout (KO) mice, despite unopposed ACE activity and elevated Ang II levels [47]. By comparison, ACE KO mice show abnormal tubuloglomerular feedback responses and display a number of alterations in kidney morphology [48]. We have reported that ACE2 KO mice are more sensitive to cigarette smoke and PM2.5 exposure [49,50]; additionally, aristolochic acid I (AAI)-induced nephropathy was more severe in ACE2 KO mice than in wild-type mice [51]. The effects of RAS regulation, ACE2 and kidney diseases in the era of COVID-19 have also been discussed. However, no clear conclusion could be reached [52,53].

In this study, as expected, captopril inhibited ACE activity but increased chymase expression levels. However, chymostatin inhibited chymase and induced ACE expression (Figure 5). Interestingly, both drugs decreased Ang II expression levels in the kidney. Furthermore, levels of p-ERK1/2, p-JNK, and p-p38 were reduced by treatment with either captopril or chymostatin. These findings support the assumption that intrarenal Ang II can be produced by either of two pathways: a traditional ACE-dependent mechanism or an ACE-independent pathway, such as that mediated by chymase [54,55]. In our study, combined treatment with captopril and chymostatin did not seem to be more effective than a single-drug therapy. However, data regarding the definitive role of chymase in the pathogenesis of DN are rare [51,56], and these two drugs did not induce a full reversal of or even a halt in the deterioration of renal function. Other RAS inhibitors, such as angiotensin II receptor blockers (ARBs), renin inhibitors and even chymase inhibitors (CI), are now considered key drugs for the treatment of hypertensive patients with kidney disease [57]. This study focused only on the use of captopril (ACEI) and chymostatin (CI). Both of these inhibitors work directly on Ang II production. Further studies could explore the use of combination treatments for DN.

According to our results, DM and HFD induced imbalances in RAS, as well as severe signaling activation. RAS participates in the cellular inflammatory and oxidative responses of the kidney and heart [58]. It is well known that the main signaling pathways in the RAS system are ERK1/2, JNK1/2, and p38 in MAPK [59]. The activation of MAPK signaling was found under high-glucose conditions/hyperglycemia in both in vivo and in vitro studies; the inhibition of phosphorylated ERK (p-ERK), p-JNK, and p-p38 could slow DM progression and decrease organ injury [60,61,62,63]. These three signaling pathways are involved in regulating a wide range of cellular processes, including cell proliferation, differentiation, inflammation, and immune responses. Here, the DN mechanisms were found to be similar between HFD-fed and abnormal RAS db/db and ACE2 KO mice, with both showing the activation of the MAPK signaling pathway via increased p-JNK and p-p38 expression levels in the kidney (Figure 4). The exact relationship between the MAPK pathway and RAS in the pathophysiology of DM disease remains to be explored, but this study suggests that the MAPK signaling pathway is important in the modulation of ACE and ACE2 expression and that the inhibition of MAPK signaling could effectively preclude high-glucose-induced kidney damage in tubular cells [64].

Tang et al. [65] reported that p-JNK and p-p38 expression are increased by renal injury. Some researchers have also reported that inhibition of p-JNK expression promotes positive energy balance, adiposity, metabolic inflammation, and insulin resistance [66]. Ijaz et al. [61] observed that p-JNK expression is increased in the glomeruli but shows no change in podocytes in the early phases of DN in db/db mice. They also concluded that JNK signaling activation is important in the development of albuminuria. Regarding p-p38 expression, Adhikary et al. [60] reported that p38 signaling might play an important role in the maintenance of normal tubular function, as p-p38 was identified in some epithelial cells within normal kidney cortical tubules. In db/db mice, p-p38 expression levels continuously increased with increasing age in glomerular cells, whereas they decreased in tubular cells [64]. In db/db and ACE2 KO mice, DM progression was shown to be more aggravated by increased Ang II generation compared to db/db mice, although renal chymase activity was shown to be similar between db/db and ACE2 KO mice and db/db mice. Thus, db/db and ACE2 KO mice are considered a better model of the effects of RAS on the progression of DN [67]. It is important to note that the rodent models used to study DN have certain limitations in replicating disease progression and accurately representing human disease. Therefore, alternatives such as nonhuman primate models and studies involving human patients with DN should be considered for further investigations. Additionally, sex differences have been reported in the structure, function, and development of DN [68]. To avoid interference from sex hormones, male mice were used in this study. However, we agree that researchers should avoid gender bias. Some studies have shown no sex-specific differences in renal structural and functional damage in db/db mice [69], but it is not known whether there are sex differences in renal injury and renal dysfunction in db/db and ACE2 KO mice.

Because RAS activation in the kidneys is a major mediator of renal damage in DN [70], small-molecule inhibitors targeting the RAS are used for the treatment of DN [17]. ACEI and CI, which were used in RAS inhibition in our experiment, have been used to treat HFD-induced kidney injury. ACEI and captopril have been widely investigated in many types of research [71,72]; clinically, they have been shown to slow DM while having no effects on blood glucose and lipid levels; they act by reducing albuminuria and alleviating fibrosis in the kidneys [72]. These data are consistent with our findings. Compared with ACEI, less work has been done on the use of CI to treat DM. A recent study showed that the inhibition of chymase reduced hyperlipidemia, hypertension, inflammation, and fibrosis without influencing blood glucose, lipid levels or blood pressure in animal models of DM [73]. Zhang et al. [74] demonstrated that CI treatment improves DN by decreasing KW/HW and UACR, as well as the expression of TGF-β and fibronectin mRNA in the kidney. Additionally, our study showed that CI treatment reduced serum cholesterol and triglyceride, a finding that differs from that of Takai and Jin [73]. In our research, chymostatin treatment lowered KW/HW and albuminuria, similarly to ACEI treatment. Additionally, chymostatin treatment reduced blood glucose and blood lipid, and it was more effective than ACEI treatment. This result may confirm that ACE and chymase play distinct roles in DM and that the inhibition of chymase might have other effects during therapy for hyperlipidemia and hyperglycemia.

## 4. Materials and Methods

### 4.1. Chemicals and Reagents

The primary antibodies against p-ERK1/2, p-JNK, p-p38, ACE, chymase, Ang II and β-actin, as well as peroxidase (HRP)-conjugated secondary antibodies against goat anti-rabbit IgG and goat anti-mouse IgG, were obtained from Genetex (Irvine, CA, USA) or Cell Signaling Technology (Beverly, MA, USA). Chemiluminescence substrates were visualized using enhanced chemiluminescence detection (Western Lightning Plus-ECL; PerkinElmer, Boston, MA, USA) and a luminescence imaging system (LAS-3000; Fuji Film, Stamford, CT, USA). Polyvinylidene fluoride membranes (PVDF) were purchased from Merck (Kenilworth, NJ, USA), and captopril (ACEI; #C4042) and chymostatin (CI; #C7268) were bought from Sigma-Aldrich (St. Louis, MO, USA).

### 4.2. Experimental Animals

All experiments were carried out in accordance with institutional guidelines and with the approval of the Institutional Animal Care and Use Committee (IACUC) at National Chiao Tung University (NCTU). The vivarium was maintained at 23 °C on a 12 h light/dark cycle with lights turned off at 7 p.m. The number of mice used in this study was minimized to the best of our ability.

#### 4.2.1. db/db and ACE2 Double-Gene-Knockout Mice

ACE2 KO (B6; 129S5-Ace2tm1/Lex/Mmcd) mice were obtained at 7 weeks of age from the Mutant Mouse Regional Resource Centers (MMRRC) and bred in the NLAC. All animal experiments conformed to the “Guide for the Care and Use of Laboratory Animals published by National Institutes of Health” (NIH Publication No. 85-23, revised 1996) and were approved by the Animal Welfare Committee of National Chiao Tung University (NCTU-IACUC-110002). The number of mice used in this study was minimized to the best of our abilities. ACE2 KO mice have been investigated in recent studies in our laboratory [13,50,67]. ACE2 KO mice and db heterozygous (db/+) mice are assumed to be more susceptible to diet-induced DM, and the results show that ACE2 plays a crucial role in DN. Thus, our lab has created breeding strategies to obtain a new strain of db/db mice without ACE2 [67]. In brief, db/db and ACE2 double-gene-knockout mice were created via the mating of male db heterozygous mice with female ACE2 KO mice.

#### 4.2.2. Animal Model of Accelerating DN with Age and HFD Feeding

Thirty male db/db and ACE2 KO mice at 7 weeks of age were used in this study. After 1 week of acclimatization, all mice were randomly divided into five groups. These groups were as follows: (1) 8 weeks old, ND; (2) 12 weeks old, ND; (3) 12 weeks old, HFD; (4) 12 weeks old, HFD + captopril (ACEI); and (5) 12 weeks old, HFD + chymostatin (chymase inhibitor; CI) (Figure 2). Before the mice were sacrificed, they were anesthetized using avertin (250 mg/kg). To study HFD-induced DN and the function of RAS inhibition, db/db and ACE2 KO mice were fed with 10 kcal% of normal diet (ND; ORIENTAL YEAST Co., Tokyo, Japan) or 60 kcal% of an HFD (Research Diets Inc., New Brunswick, NJ, Canada) for 4 weeks; after 2 weeks on the HFD, the mice also received ACEI (captopril) or CI (chymostatin) at a dose of 10 mg/kg twice daily, 7 times, in filtered phosphate buffered saline (for captopril) or DMSO (for chymostatin), via intraperitoneal injection (i.p.). he other group received an equal volume of PBS.

### 4.3. Sample Collection and Biochemical Determinations

The body weight, food intake, and water intake of db/db and ACE2 KO mice were recorded weekly during the experiments. Serum samples were collected by direct cardiac puncture and stored at −25 °C, and urine samples from each group were collected on the day before sacrifice using metabolic cages (Tecniplast, Buguggiate, Italy) for 24 h (6 a.m.–6 p.m.). The samples were stored at −20 °C after centrifugation at 3000 rpm for 15 min. After the severity of their pathology was determined, the mice were anesthetized with avertin (250 mg/kg) and whole blood was collected by cardiac puncture.

After the collection of the whole blood, we allowed the blood to clot by leaving it undisturbed on ice for 15–30 min, then removed the blood clot by centrifugation at 3000 rpm for 15 min at 4 °C to acquire serum (supernatants). The kidney and heart were excised after perfusion with 0.9% NaCl solution and kept on ice for the next steps (tissue homogenate and histological determination). For the histological evaluation, the kidney samples were fixed with 10% formaldehyde for 24 h. The stained sections were photographed using a digital camera mounted on a microscope.

To evaluate renal function and blood lipid levels in the mice, the levels of serum triglycerides (TG), total cholesterol (TCHO), and creatinine (CRE) were measured using a clinical chemical analyzer (#DRI-CHEM 3500; Fujifilm Medical, Tokyo, Japan). The urine albumin (ALB) and CRE samples were evaluated using the same analyzer.

### 4.4. Protein Extraction

Organ samples were obtained from the db/db and ACE2 KO mice and were used for further analyses, as described. After the mice were sacrificed, the organ samples were collected, weighed, and homogenized 3 to 5 times in 1 mL lysis buffer PRO-PREPTM Protein Extraction Solution (iNtRON Biotechnology, Kyungki-Do, Republic of Korea). The homogenized samples were incubated on ice for about 20 min; after incubation, the samples were transferred to a 1.5 mL Eppendorf tube, then sonicated 3 times for 5 s each on ice, with an interval of 10 s, using an ultrasonic processor. Finally, the samples were centrifuged for 10 min at 13,000 rpm at 4 °C (Biofuge primoR; Sorvall, Osterode, Germany) and the supernatants were transferred to clean vials. The total amounts of protein in homogeneous supernatants were measured via a Bradford dye-binding assay (Bio-Rad Laboratories, Hercules, CA, USA) with bovine serum albumin as the standard. The supernatants were aliquoted and stored at −80 °C until further use.

### 4.5. Western Blot Assay

Equal amounts of proteins (25 µg) were resolved in 10% SDS-PAGE gel and transferred to the polyvinyl difluoride (PVDF) membrane. After blocking with 5% non-fat milk, the membranes were incubated overnight with antibodies against p-ERK1/2, p-JNK, p-p38, ACE, chymase, Ang II, and β-actin, then subsequently incubated with peroxidase (HRP)-conjugated secondary antibodies at a dilution of 1:2000–4000 for 1 h at room temperature. Finally, the immunocomplex was visualized using enhanced chemiluminescence detection and the membranes were exposed to X-ray film. The bands on the film were detected at the anticipated locations for their size. Band intensity was quantified using the ImageJ program (NIH, Bethesda, MD, USA) and normalized to the β-actin level.

### 4.6. Histological Determination

Kidney tissue was excised from db/db and ACE2 KO mice, soaked in 10% formaldehyde overnight, embedded in paraffin, and cut into 6–8 μm-thick sections on acid-pretreated glass slides, then deparaffinized. It was then rehydrated and stained with hematoxylin and eosin (H&E), Masson’s trichrome (Masson’s) or Periodic Acid–Schiff (PAS) solution using standard pathology procedures [67]. The stained pathological sections were photographed with a digital camera mounted on a microscope.

To assess glomerular or tubular injury by metrics such as lymphocyte infiltration, glomerular hypertrophy, collagen deposition, glomerular mesangial expansion, brush-border loss in the proximal tubule, and tubular atrophy, a computerized microscope equipped with a high-resolution video camera (BX 51; Olympus, Tokyo, Japan) and the ImageJ program were used for quantitative and morphometric analyses., Five histological images were randomly selected in each group. Positive signals within the selected images were highlighted and measured, with the positive area then quantified as a percentage of the entire glomerulus or tubule.

### 4.7. Statistical Analysis

All parameters are expressed as the mean ± standard deviation (SD). Statistical significance between the two groups was assessed using Student’s *t*-test, and differences between groups were assessed using one-way analysis of variance (ANOVA). A *p*-value less than 0.05 is considered to indicate a statistically significant difference.

## 5. Conclusions

The present study represents the first use of the db/db mouse model to prove that ACE2 deficiency, i.e., ACE2 KO, not only promotes HFD-induced hyperglycemia and nephropathy, but also causes MAPK signaling activation. In the in vivo study, db/db and ACE2 KO mice were successfully used as a model to investigate the roles of ACE and chymase in RAS. We observed the effects of 4 weeks of feeding the HFD on db/db and ACE2 KO mice, concluding that it aggravated DN progression through the over-activation of the MAPK and RAS signaling pathways. Our study also confirmed that both ACE-dependent and chymase-dependent pathways play crucial roles in the generation of Ang II independent of the ACE2-Ang 1-7/Mas axis. Both captopril and chymostatin halt the development of HFD-induced kidney damage. Blocking the activity of the RAS using the currently available agents is only partially successful in patients with renal and heart diseases. However, the side effects can be potentially damaging and still need to be explored. Future studies must consider the large array of components in the RAS cascade, which represent many potential targets for inhibition.

## Figures and Tables

**Figure 2 ijms-25-00329-f002:**
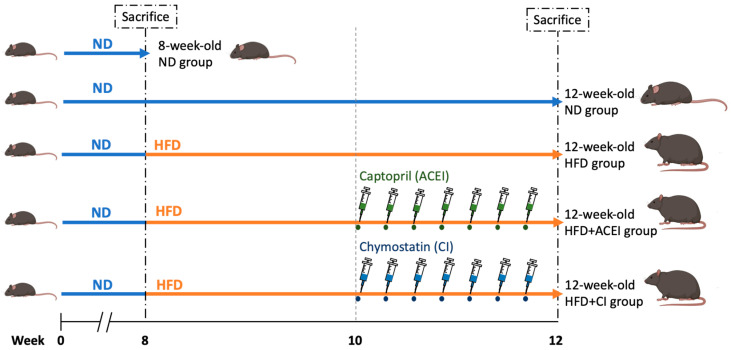
Experimental design of feeding the HFD and therapeutic treatments in db/db and ACE2 KO mice. The mice were divided into five groups. One group of eight-week-old mice was sacrificed, and we sampled the blood and tissues for assay as the Control group. Next, db/db and ACE2 KO mice were fed a high-fat diet (HFD) for four weeks. After two weeks on the HFD, captopril (ACE inhibitor, ACEI) or chymostatin (chymase inhibitor, CI) were administered in a twice-daily dose of 10 mg/kg via intraperitoneal injection (i.p.). These mice were sacrificed at 12 weeks old, and we sampled the blood and tissues for further assay.

**Figure 3 ijms-25-00329-f003:**
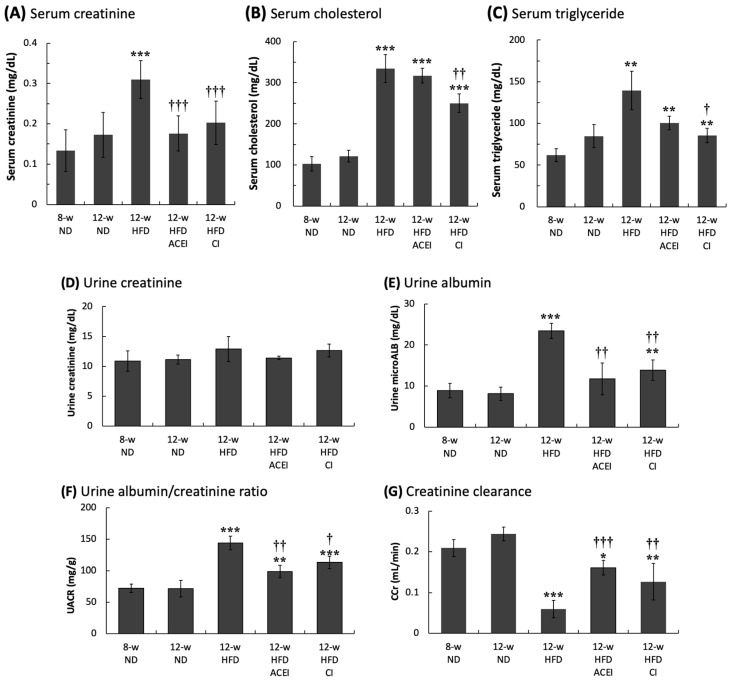
The serum and urinary biochemical parameters of db/db and ACE2 KO mice. Serum creatinine (**A**), total cholesterol (**B**) and triglyceride (**C**) were significantly increased in the HFD group. Both captopril (ACEI) and chymostatin (CI) treatments improved serum creatinine, but only chymostatin lowered the blood lipid level. Urine creatinine (**D**) showed no marked difference across groups. The high-fat diet (HFD) significantly increased the urine albumin (**E**) and urine albumin/creatinine ratio (UACR) (**F**) and significantly decreased the creatinine clearance (CCr) (**G**); these parameters were measured as a test of kidney function. Administration of ACEI or CI slowed the loss of kidney function. All parameters are expressed as the mean ± SD (*n* = 6) from each group. *, ** and *** indicate *p* < 0.05, *p* < 0.01 and *p* < 0.001, respectively, compared with the 12-week-old ND group. ^†^, ^††^ and ^†††^ indicate *p* < 0.05, *p* < 0.01 and *p* < 0.001, respectively, compared with the 12-week-old HFD group.

**Figure 4 ijms-25-00329-f004:**
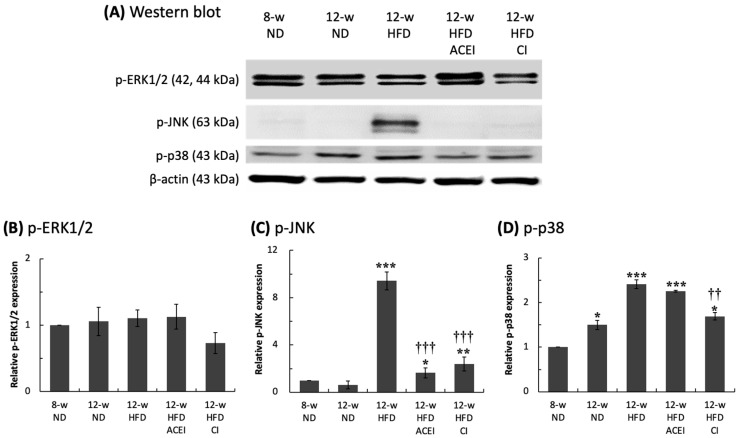
Relative expression of renal MAPK-related protein in db/db and ACE2 KO mice. The kidneys were sampled and protein was extracted for Western blotting assays (**A**). High-fat-diet (HFD) feeding did not affect the p-ERK1/2 level (**B**); however, significantly increased p-JNK (**C**) and p-p38 (**D**) levels were detected. Administration with captopril (ACEI) or chymostatin (CI) could slow down the abnormal MAPK activation. All parameters were expressed as the mean ± SD (*n* = 6) for each group. *****, ****** and ******* indicate *p* < 0.05, *p* < 0.01 and *p* < 0.001, compared with the 8-w (8-week-old) ND group, respectively. ^††^ and ^†††^ indicate *p* < 0.01 and *p* < 0.001, compared with the 12-w (12-week-old) HFD group, respectively.

**Figure 5 ijms-25-00329-f005:**
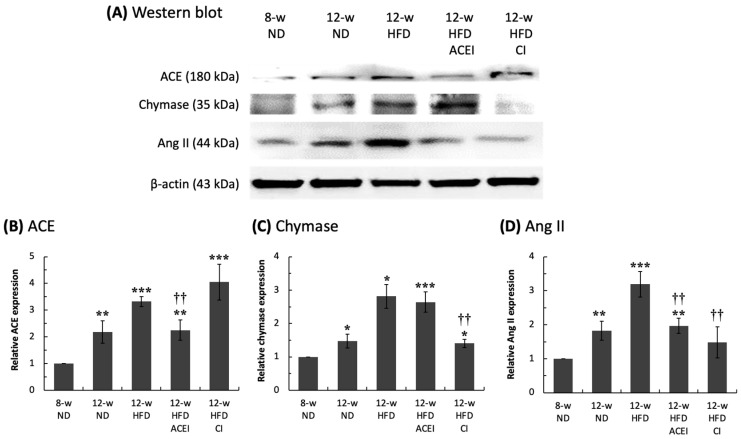
Relative levels of renal–RAS components in db/db and ACE2 KO mice. The kidneys were sampled, and proteins were extracted for Western blot assays (**A**). Feeding a high-fat diet (HFD) significantly accelerated abnormal activity in the renin–angiotensin system (RAS) by increasing ACE (**B**), chymase (**C**), and Ang II (**D**) levels. Administration of captopril (ACEI) and chymostatin significantly ameliorated the HFD-induced increase in Ang II levels. All parameters are expressed as the mean ± SD (*n* = 6) from each group. *****, ****** and ******* indicate *p* < 0.05, *p* < 0.01 and *p* < 0.001, respectively, compared with the 8-w (8-week-old) ND group. ^††^ indicates *p* < 0.01 compared with the 12-w (12-week-old) HFD group.

**Figure 6 ijms-25-00329-f006:**
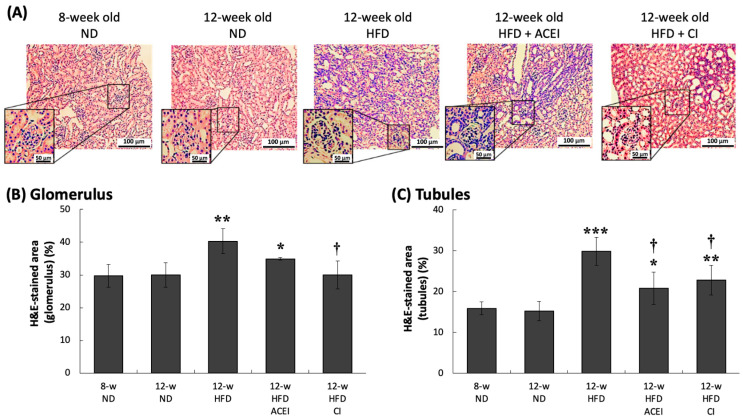
Hematoxylin and eosin (H&E)-stained kidney sections of db/db and ACE2 KO mice. Kidney sections from the mice were stained with H&E and then photographed for pathologic examination at 100× magnification (**A**). Captopril (ACEI) and chymostatin (CI) treatments decreased the high-fat-diet (HFD)-induced lymphocytic infiltration in the tubules (**B**,**C**). Glomerular hypertrophy was observed in the 12-week-old HFD group. All parameters are expressed as the mean ± SD for each group. *****, ****** and ******* indicate *p* < 0.05, *p* < 0.01 and *p* < 0.001, respectively, compared with the 8-w (8-week-old) ND group. ^†^ indicates *p* < 0.05 compared with the 12-w (12-week-old) HFD group (*n* = 3 for each group).

**Figure 7 ijms-25-00329-f007:**
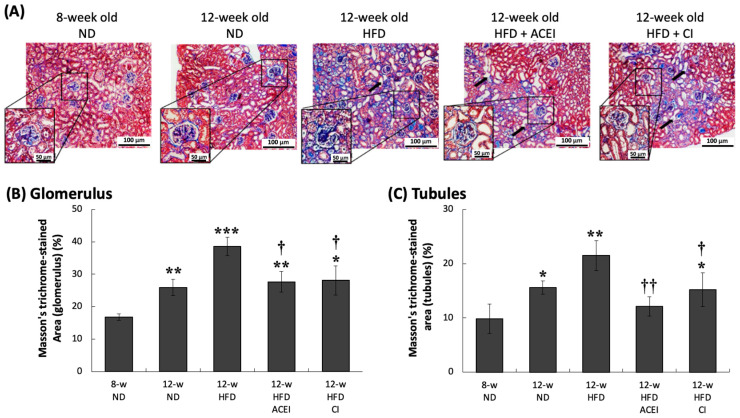
Masson’s trichrome-stained kidney sections of db/db and ACE2 KO mice. Kidney sections from the mice were stained with Masson’s trichrome and then photographed for pathologic verification at 100× magnification (**A**). Both the captopril (ACEI) and chymostatin (CI) treatments reduced the HFD-induced collagen accumulation in the glomerulus and tubules. Slightly greater glomerular injury was found in 12-w (12-week-old) ND groups (**B**,**C**). All HFD-fed groups showed greater evidence of missing or ruptured brush border and glomerular hypertrophy. All parameters are expressed as the mean ± SD from each group. *****, ****** and ******* indicate *p* < 0.05, *p* < 0.01 and *p* < 0.001, respectively, compared with the 8-w (8-week-old) ND group. ^†^ and ^††^ indicate *p* < 0.05 and *p* < 0.01, respectively, compared with the 12-w (12-week-old) HFD group (*n* = 3 for each group).

**Figure 8 ijms-25-00329-f008:**
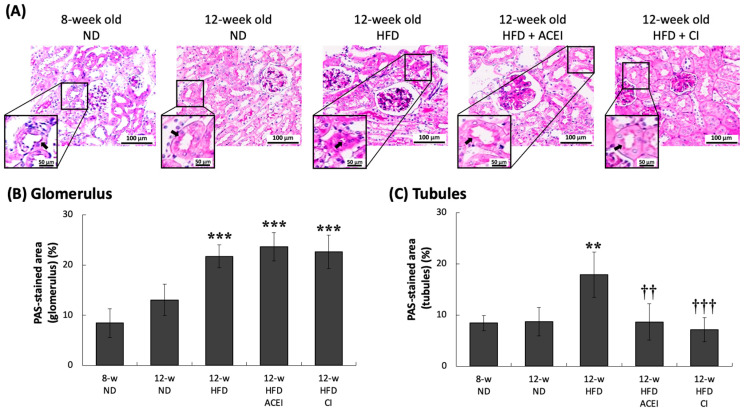
PAS-stained kidney sections of db/db and ACE2 KO mice. Kidney sections from the mice were stained with PAS and then photographed for pathologic assessment at 200× magnification (**A**). Accumulations of PAS-positive matrix in the mesangium and severe tubular atrophy were observed in all of the HFD-fed groups. Administrating captopril (ACEI) or chymostatin (CI) ameliorated tubular atrophy but did not decrease mesangial matrix expansion (**B**,**C**). All parameters are expressed as the mean ± SD from each group. ****** and ******* indicate *p* < 0.01 and *p* < 0.001, respectively, compared with the 8-w (8-week-old) ND group. ^††^ and ^†††^ indicate *p* < 0.01 and *p* < 0.001, respectively, compared with the 12-w (12-week-old) HFD group (*n* = 3 for each group).

**Table 1 ijms-25-00329-t001:** The physical characteristics of db/db and ACE2 KO mice in each experimental group.

	8-Week-Old ND	12-Week-Old ND	12-Week-Old HFD	12-Week-Old HFD + ACEI	12-Week-Old HFD + CI
**Body weight** **(BW) (g)**	45.4 ± 3.3	51.8 ± 5.1	64.8 ± 7.5 *	59.6 ± 4.8	59.1 ± 3.6
**Blood glucose** **(mg/dL)**	256 ± 24	368 ± 20 ***	620 ± 42 ***	589 ± 44 ***	499 ± 43 *** ^††^
**Kidney weight** **(KW) (g)**	0.305 ± 0.037	0.412 ± 0.065 *	0.545 ± 0.069 ***	0.404 ± 0.071 * ^†^	0.348 ± 0.065 ^††^
**KW/BW (%)**	6.54 ± 0.69	7.59 ± 0.56	7.82 ± 0.18 *	7.40 ± 0.59	6.56 ± 0.81 ^†^

Mice were fed a normal diet (ND; 10 kcal%) or high-fat diet (HFD; 60 kcal%) from 8 weeks old and were administered captopril (ACE inhibitor, 10 mg/kg/2 days, 7 times, i.p.) or chymostatin (CI, 10 mg/kg/2 days, 7 times, i.p.) from 10 to 12 weeks old. All parameters are expressed as the mean ± SD (*n* = 6) from each group. * and *** indicate *p* < 0.05 and *p* < 0.001, respectively, compared with the 8-week-old ND group. ^†^ and ^††^ indicate *p* < 0.05 and *p* < 0.01, respectively compared with the 12-week-old HFD group.

## Data Availability

The data presented in this study are available upon request from the corresponding author.

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
