# Peer review of "Studying the Roles of the Renin–Angiotensin System in Accelerating the Disease of High-Fat-Diet-Induced Diabetic Nephropathy in a db/db and ACE2 Double-Gene-Knockout Mouse Model"

_ijms, 2023, doi:10.3390/ijms25010329_

Round 1

Reviewer 1 Report

Comments and Suggestions for Authors

Dear authors, the use of a db/db and ACE2 knockout mouse model offers a unique perspective on the progression of DN, and your findings regarding the potential therapeutic roles of ACE inhibitors and chymase inhibitors are noteworthy.

Here are some sections that could use improvement:

Introduction: The manuscript's introduction effectively contextualizes the study in diabetic nephropathy research but can benefit from a more concise summary of relevant prior findings. A clearer articulation of specific study objectives or hypotheses would enhance focus and reader understanding.

Methods: The methods section provides a solid framework for experimental procedures, particularly in describing the db/db and ACE2 knockout mouse model and dietary interventions.

Results: The results section is comprehensive, presenting data effectively with various parameters. To enhance impact, highlight the most significant findings that directly address research questions. A more focused presentation through selective emphasis or summarization would increase reader-friendliness, some figures could be compacted in a single one within sections.

Discussion: A limitations sections would add to the paper, several potential limitations that are not explicitly addressed. Acknowledging and discussing these limitations would enhance the credibility and depth of your study, some examples:

1.     Model Specificity: While the db/db and ACE2 knockout mouse model is valuable for studying diabetic nephropathy (DN), it may not fully replicate the human condition. Differences in disease progression, metabolic responses, and genetic background between mice and humans can impact the translatability of findings.    

2.     Scope of RAS Inhibitors Studied: The study focuses on specific RAS inhibitors, but there are many other components and pathways within the RAS that might also contribute to DN pathophysiology. A broader examination of these components could provide a more comprehensive understanding.

3.     Sample Size and Statistical Power: If the sample size is small or not mentioned, this could limit the statistical power of the study and the ability to generalize findings.

Comments on the Quality of English Language

The English quality of the manuscript needs improvement to enhance clarity and professionalism. Grammatical errors, simplifying complex sentences, ensuring consistent terminology, and correcting typographical errors need to be adressed. A review by a native English speaker specialized in scientific writing is recommended to ensure your valuable research is clearly and effectively presented.

Some examples in the abstract:

  • "severe disease progression of ND" could be clearer as "severe progression of diabetic nephropathy (DN)."
  • "HFD-fed db/db & ACE2 KO mice is" could be revised to "HFD-fed db/db & ACE2 KO mice are."
  •  

Acronyms like DN are mentioned as ND or similar through the paper.

Author Response

Dear reviewer:

Thank you very much for your reviewing process of my manuscript “Studying the roles of renin angiotensin system in the accelerating disease of high-fat-diet-induced diabetic nephropathy in a db/db and ACE2 double gene knockout mouse model (ijms-2747754)”. The criticisms raised by the reviewer were extremely helpful and have been fully integrated into this the revised submission. We deeply appreciate the reviewers’ detailed comments to improve the readability of the manuscript; each of their points has been addressed. Revised portions are highlighted in red in the revised manuscript.

The followings are our point-to-point responses to the comments:

Elucidation for Reviewer 1:

Comment:

Dear authors, the use of a db/db and ACE2 knockout mouse model offers a unique perspective on the progression of DN, and your findings regarding the potential therapeutic roles of ACE inhibitors and chymase inhibitors are noteworthy.

Here are some sections that could use improvement:

Introduction: The manuscript's introduction effectively contextualizes the study in diabetic nephropathy research but can benefit from a more concise summary of relevant prior findings. A clearer articulation of specific study objectives or hypotheses would enhance focus and reader understanding.

Response: Thanks for your efforts and time to review our manuscript. To articulate the specific study objectives, we revised the statements in the Introduction section as follows:

The pathological mechanisms of DN are intricate and complex. It is lack of alternated animal model to simulate all the pathological features of human DN, especially the characteristic pathological changes in the progressive and late stages of DN. Thus, the purposes of this study are the exploration of DN progression in db/db and ACE2 KO mice and enhancing the understanding of the role of and changes in RAS factors after HFD feeding. (in the revised manuscript v.3, lines 104-109)

Methods: The methods section provides a solid framework for experimental procedures, particularly in describing the db/db and ACE2 knockout mouse model and dietary interventions.

Response: Many thanks for your comment and agreement.

Results: The results section is comprehensive, presenting data effectively with various parameters. To enhance impact, highlight the most significant findings that directly address research questions. A more focused presentation through selective emphasis or summarization would increase reader-friendliness, some figures could be compacted in a single one within sections.

Response: Thank you for the pertinent comment. The Figure 3 (original copy) was deleted and Figure 4 and 5 had been merged (revised Figure 3) for condensing the focus of data presentation and increasing reader-friendliness.

Discussion: A limitations sections would add to the paper, several potential limitations that are not explicitly addressed. Acknowledging and discussing these limitations would enhance the credibility and depth of your study, some examples:

  1. Model Specificity: While the db/db and ACE2 knockout mouse model is valuable for studying diabetic nephropathy (DN), it may not fully replicate the human condition. Differences in disease progression, metabolic responses, and genetic background between mice and humans can impact the translatability of findings.

Response: We sincerely appreciate and agree with the reviewer’s comment. It is important to note that the mouse models used to study DN have certain limitations in replicating disease progression and accurately representing the human condition. Therefore, we revised the manuscript in the Discussion section as follows:

It is important to note that the rodent models used to study DN have certain limitations in replicating disease progression and accurately representing the human condition. Therefore, alternative options such as nonhuman primate models and studies involving human patients with DN should be considered for further investigations.” (in the revised manuscript v.3, lines 418-422)

  1. Scope of RAS Inhibitors Studied: The study focuses on specific RAS inhibitors, but there are many other components and pathways within the RAS that might also contribute to DN pathophysiology. A broader examination of these components could provide a more comprehensive understanding.

Response: Many thanks for your comments. There are a lot of RAS inhibitors, such as angiotensin-converting enzyme inhibitors (ACEI), angiotensin II receptor blockers (ARBs), renin inhibitors and even chymase inhibitors (CI), that are now positioned as key drugs for DN. This present study only focused on the use of Captopril (ACEI) and chymostatin (CI), both inhibitors directly work on the Ang II generation. To more comprehensively understand DN therapy, we discussed other studies about RAS inhibitors in the Discussion section.

“Other RAS inhibitors such as angiotensin II receptor blockers (ARBs), renin inhibitors and even chymase inhibitors (CI), are now positioned as key drugs for the treatment of hypertensive patients with kidney disease. The present study only focused on the use of Captopril (ACEI) and chymostatin (CI), both inhibitors directly work on the Ang II generation. Further studies could explore the combination treatments for DN.” (in the revised manuscript v.3, lines 382-387)

  1. Sample Size and Statistical Power: If the sample size is small or not mentioned, this could limit the statistical power of the study and the ability to generalize findings.

Response: Thanks for the reviewer’s suggestion. We totally agree that the scientific value of studies with large sample sizes is higher than that of small ones. However, the 3R principles (Replacement, Reduction, and Refinement) indicate that methods should avoid or replace the use of animals, and minimize the number of animals used consistent with scientific aims in research. According to the 3R principles, the numbers of mice used in this study were minimized to the best of our abilities.

Comments on the Quality of English Language:

The English quality of the manuscript needs improvement to enhance clarity and professionalism. Grammatical errors, simplifying complex sentences, ensuring consistent terminology, and correcting typographical errors need to be adressed. A review by a native English speaker specialized in scientific writing is recommended to ensure your valuable research is clearly and effectively presented.

Some examples in the abstract:

  • "severe disease progression of ND" could be clearer as "severe progression of diabetic nephropathy (DN)."
  • "HFD-fed db/db & ACE2 KO mice is" could be revised to "HFD-fed db/db & ACE2 KO mice are."

Acronyms like DN are mentioned as ND or similar through the paper.

Response: Thanks for the reviewer’s reminder, we checked the grammatical and spelling mistakes, and corrected it throughout the revised manuscript. We hope you will find our revised manuscript has been seriously and carefully revised and the revised version has been edited by the professional scientific editing service of MDPI partner. The English editing certificate was shown in the bottom in this response letter.

We hope you will find our revised manuscript has been seriously and carefully revised and the revised version has been edited by the professional scientific editing service of MDPI partner. The English editing certificate was shown in the bottom in this response letter. We hope the revised manuscript is satisfactory and suitable for publication in your journal, International Journal of Molecular Sciences.

With best regard

Chih-Sheng Lin, Ph.D.

Distinguished Professor

Department of Biological Science and Technology

National Yang Ming Chiao Tung University

Bio-ICT building Rm.722,

Address: No.75 Po-Ai Street, Hsinchu 30068, Taiwan

Tel.: +886-3-5131338

E-mail: lincs@nycu.edu.tw

The English-Editing-Certificate of ijms-2747754

Reviewer 2 Report

Comments and Suggestions for Authors

Major Comments:

The manuscript is well written. However, the authors are requested to address the following queries:

1.   What is the novelty of the study?

2.   The authors should mention unabbreviated form before using acronym.

3.   Why did the authors choose male mice in their study? Is there any specific reason? They should provide the data for female too to avoid the potential gender specific bias in their result.

4.   Do the authors have any data regarding male vs. female mice?

5.   The authors should check the levels of ECM proteins and MMPs as these are very important for fibrotic renal disfunction in DN. In recent studies (PMID: 34680110, 36522378) showed the role of collagen, fibronectin, MMPs and EMT in renal fibrosis in diabetic nephropathy. The authors may enlighten these important aspects in the ‘Discussion’ of their manuscript.

6.     Why didn’t the authors consider checking KIM-1 in their experimental mice?

Minor Comments:

1.     Line-34: ‘ND’??

2.     The authors should thoroughly check the manuscript for the potential grammatical errors.

Comments on the Quality of English Language

Minor editing of English language required.

Author Response

Dear reviewer:

Thank you very much for your reviewing process of my manuscript “Studying the roles of renin angiotensin system in the accelerating disease of high-fat-diet-induced diabetic nephropathy in a db/db and ACE2 double gene knockout mouse model (ijms-2747754)”. The criticisms raised by the reviewer were extremely helpful and have been fully integrated into this the revised submission. We deeply appreciate the reviewers’ detailed comments to improve the readability of the manuscript; each of their points has been addressed. Revised portions are highlighted in red in the revised manuscript.

The followings are our point-to-point responses to the comments:

Elucidation for Reviewer 2:

Major Comments:

The manuscript is well written. However, the authors are requested to address the following queries:

  1. What is the novelty of the study?

Response: Many thanks for your comment. The pathological mechanisms of DN were intricate and complex, involved many factors such as genetics, oxidative stress and inflammation. It is lack of animal had yet been able to simulate all the pathological features of human DN, especially the characteristic pathological changes in the progressive and late stages of DN. Thus, the purposes of this study are the exploration of DN progression in db/db and ACE2 KO mice and enhancing the understanding of the role of and changes in RAS factors after HFD feeding. That is, the genetic deficiency of ACE2 plus short-term HFD feeding in db/db mice is used to accelerate DN progression in the present study.

To articulate the specific study objectives, we added the statements in the Introduction section as follows:

The pathological mechanisms of DN were intricate and complex. There had been no alternated animal model to simulate all the pathological features of human DN, especially the characteristic pathological changes in the progressive and late stages of DN. Thus, the purposes of this study are the exploration of DN progression in db/db and ACE2 KO mice and enhancing the understanding of the role of and changes in RAS factors after HFD feeding. (in the revised manuscript v.3, lines 104-109)

  1. The authors should mention unabbreviated form before using acronym.

Response: Thanks for the reviewer’s reminder. We checked all the abbreviations had been defined when the first mention.

  1. Why did the authors choose male mice in their study? Is there any specific reason? They should provide the data for female too to avoid the potential gender specific bias in their result.
  2. Do the authors have any data regarding male vs. female mice?

Response: I do deeply appreciate for your comments. Unfortunately, we did not have the data regarding male vs. female mice. It has been reported that sex differences in the structure, function and development of DN (Clotet et al., 2016). To avoid the interference of sex hormone, there are male mice used in this study. However, we totally agreed the result should be avoid the potential gender specific bias. Some study shown that no sex difference in renal structural and functional damage in db/db mice (Ma et al., 2019). It is not known whether there are sex differences in renal injury and renal dysfunction in db/db and ACE2 KO mice, we discussed the gender aspect in the Discussion section as follows:

“Besides, sex differences have been reported in the structure, function and development of DN [68]. To avoid the interference of sex hormone, there were male mice used in this study. However, we totally agreed the result should be avoided the potential gender specific bias. Some study shown that no sex difference in renal structural and functional damage in db/db mice [69]. It is not known whether there are sex differences in renal injury and renal dysfunction in db/db and ACE2 KO mice” (in the revised manuscript v.3, lines 422-427)

The two references have been cited in the revised manuscript v.3.

  1. Clotet S, Riera M, Pascual J, Soler MJ. RAS and sex differences in diabetic nephropathy. 2016. Am J Physiol Renal Physiol310: F945–F957, doi: 10.1152/ajprenal.00292.2015.
  2. Ma Y, Li W, Yazdizadeh Shotorbani P, Dubansky BH, Huang L, Chaudhari S, Wu P, Wang LA, Ryou MG, Zhou Z, Ma R. Comparison of diabetic nephropathy between male and female eNOS-/-db/dbmice. 2019. Am J Physiol Renal Physiol. 316(5):F889-F897. doi: 10.1152/ajprenal.00023.2019.

  1. The authors should check the levels of ECM proteins and MMPs as these are very important for fibrotic renal disfunction in DN. In recent studies (PMID: 34680110,36522378) showed the role of collagen, fibronectin, MMPs and EMT in renal fibrosis in diabetic nephropathy. The authors may enlighten these important aspects in the ‘Discussion’ of their manuscript.

Response: Thank you for your valuable suggestion and advice. ECM proteins and MMPs are important for fibrotic renal disfunction in DN. This is a major track for our further studies on the development of DN. To depth of this study, we revised the manuscript as followings:

DN is characterized by excessive extracellular matrix (ECM) accumulation leading to renal fibrosis, and matrix metalloproteinases (MMPs) as well as tissue inhibitors of metalloproteinases (TIMPs) are known to regulate synthesis and degradation of the ECM [40;41]. Kidney injury molecule-1 (KIM-1) is also markedly up-regulated in post-ischemic kidney and acute renal failure [42]. In the fibrosis pathogenesis of DN, KIM-1, MMPs and TIMPs play a crucial role, that are triggered by oxidation and inflammatory stimulation. These issues are under exploration to clarify molecular mechanisms of fibrosis pathogenesis in the db/db and ACE2 KO mice.” (in the revised manuscript v.3, lines 349-356)

The three references have been cited in the revised manuscript v.3.

  1. Juin SK, Pushpakumar S, Sen U. GYY4137 Regulates Extracellular Matrix Turnover in the Diabetic Kidney by Modulating Retinoid X Receptor Signaling. 2021. Biomolecules. 11(10):1477. doi: 10.3390/biom11101477.
  2. Juin SK, Pushpakumar S, Tyagi SC, Sen U. Glucosidase inhibitor, Nimbidiol ameliorates renal fibrosis and dysfunction in type-1 diabetes. 2022. Sci Rep. 12(1):21707. doi: 10.1038/s41598-022-25848-1.
  3. Han WK, Bailly V, Abichandani R, Thadhani R, Bonventre JV. Kidney Injury Molecule-1 (KIM-1): a novel biomarker for human renal proximal tubule injury. Kidney Int. 2002 Jul;62(1):237-44. doi: 10.1046/j.1523-1755.2002.00433.x.

  1. Why didn’t the authors consider checking KIM-1 in their experimental mice?

Response: We sincerely appreciate and agree with the reviewer’s comment. Kidney injury molecule-1(KIM-1) is markedly up-regulated in postischemic kidney and acute renal failure (Han et al, 2002). KIM-1 is increased early in human diabetes. However, we paid more attention to the late stages of DN marker related with renin angiotensin system (RAS); therefore, we didn’t analyze the state of KIM-1. According to the suggestion, KIM-1 has been briefly mentioned in the revised Discussion section as shown in Response 5.

Minor Comments:

  1. Line-34: ‘ND’??

Response: Thanks for the reviewer’s reminder. Line 34 “ND" shoud be corrected as "severe “diabetic nephropathy (DN)”. We checked the grammatical and spelling mistakes, and corrected it throughout the revised manuscript.

  1. The authors should thoroughly check the manuscript for the potential grammatical errors.

Response: We apologize the mistakes. We hope you will find our revised manuscript has been seriously and carefully revised and the revised version has been edited by the professional scientific editing service of MDPI partner. The English editing certificate was shown in the bottom in this response letter.

We hope you will find our revised manuscript has been seriously and carefully revised and the revised version has been edited by the professional scientific editing service of MDPI partner. The English editing certificate was shown in the bottom in this response letter. We hope the revised manuscript is satisfactory and suitable for publication in your journal, International Journal of Molecular Sciences.

With best regard

Chih-Sheng Lin, Ph.D.

Distinguished Professor

Department of Biological Science and Technology

National Yang Ming Chiao Tung University

Bio-ICT building Rm.722,

Address: No.75 Po-Ai Street, Hsinchu 30068, Taiwan

Tel.: +886-3-5131338

E-mail: lincs@nycu.edu.tw

The English-Editing-Certificate of ijms-2747754

Round 2

Reviewer 2 Report

Comments and Suggestions for Authors

The manuscript is now improved significantly.